# Self-Assessment of Mobility of People over 65 Years of Age

**DOI:** 10.3390/medicina57090980

**Published:** 2021-09-17

**Authors:** Pascal Martin, Alexander Martin Keppler, Paolo Alberton, Carl Neuerburg, Michael Drey, Wolfgang Böcker, Christian Kammerlander, Maximilian Michael Saller

**Affiliations:** 1Department of Orthopaedics and Trauma Surgery, Musculoskeletal University Center Munich (MUM), University Hospital, LMU Munich, 80336 Munich, Germany; pascal.martin@med.uni-muenchen.de (P.M.); alexander.keppler@med.uni-muenchen.de (A.M.K.); Paolo.Alberton@med.uni-muenchen.de (P.A.); carl.neuerburg@med.uni-muenchen.de (C.N.); wolfgang.boecker@med.uni-muenchen.de (W.B.); christian.kammerlander@med.uni-muenchen.de (C.K.); 2Department of Medicine IV, Geriatrics, University Hospital, LMU Munich, 80336 Munich, Germany; michael.drey@med.uni-muenchen.de; 3Traumahospital Styria, Graz & Kalwang, Austria

**Keywords:** older adults, mobility, self-assessment, gait speed, orthogeriatric co-management

## Abstract

Background and Objectives: Nowadays, various clinical scoring systems are used in the medical care of the elderly to assess the quality of mobility. However, people often tend to under- or overestimate themselves in many aspects. Since this can have serious consequences in their treatment and care, the aim of this study was to identify differences in the self and external assessment of mobility of persons over 65 years of age. Materials and Methods: 222 participants over 65 years of age and one external, closely-related relative or professional caregiver were interviewed by a unique study assistant using a standardized questionnaire. Participants were divided into people living in nursing homes and independent people living at home, where either the caregivers or the relatives provided the external assessment of mobility, respectively. The questionnaire included demographics, cognitive abilities (Mini Mental Status Test); fall risk (Hendrich 2 Fall Risk Model); as well as the Parker Mobility Score, Barthel Index, and EQ-5D-5L to measure mobility, activities of daily life and quality of life. In each case, the participant and the external person were asked for their assessment to the participants’ mobility situation. Statistical significance of the difference between self and external assessment was calculated with a Wilcoxon rank-sum test and assumed with a *p*-value of ≤ 0.05. Results: Self-assessment indicated a significantly higher value, when compared to an external assessment for the Parker Mobility Score for females in nursing homes (*p* ≤ 0.01), as well as for the Barthel Index for females (*p* ≤ 0.01) and males (*p* ≤ 0.01) in nursing homes. The *EQ-5D-5L* received a significantly higher self-assessment value for females (*p* ≤ 0.01) and males (*p* ≤ 0.01) living at home and females (*p* ≤ 0.01) and males (*p* ≤ 0.05) in nursing homes. Conclusions: Persons over 65 years of age tend to overestimate their level of mobility, quality of life and activities of daily life. Especially for people living in nursing homes, these scoring systems should be treated with caution due to the differences between the verbal statements. It is important to properly assess the mobility situation of elderly patients to ensure correct medical treatment and prevention of falls.

## 1. Introduction

Due to worldwide demographic changes, the number of persons over 65 years of age will continuously increase [1]. Maintaining sufficient mobility is a prerequisite for independence in old age [2]. The loss of mobility independence can lead to limitations in daily activities and thus to a reduced quality of life [2,3,4,5]. Various factors can lead to restricted mobility. In addition to physiological changes during ageing, chronic diseases and pain—especially after falls—can result in tremendous functional disorders [2,6]. The risk of falling increases continuously with age [6] and several studies have shown that 35% and 15% of persons over 65 years of age fall at least once or twice a year [7,8,9]. Furthermore, people in nursing homes fall five times more often than independent people living at home [6,9]. Falls often lead to immobility [2,8] due to serious consequences such as fractures, which often require hospitalization in up to 20% of all fall events [9]. In addition, falls are responsible for around 40% of all injury-related deaths [6]. Predictions of demographic trends indicate that the number of hospital treatments for people over 65 years of age will dramatically increase in the future [10]. As a result, the costs for the healthcare system could increase by 70% in the US until the year 2030 compared to 2002 [11]. Furthermore, dissatisfaction with the own mobility level and fear of repetitive falling causes people to lose confidence in moving safely, resulting in a loss of independence [6,9]. For this reason, there are more and more specialized centers that take special care of this fragile patient population. This integrated form of care has proven its worth, especially after falls and injuries [12]. The aim of the therapy is a rapid restoration of mobility and return to the activity level before the accident. Moreover, a sufficient and rapid full-weight bearing mobilization of patients will reduce further complications such as pulmonary or urinary tract infections [13]. Consequently, it is important to identify mobility limitations of older adults as early as possible in order to avoid falls and prevent mobility impairment [2]. It is possible to maintain the independence and functionality of older adults by implementing prevention programs at the right time [3].

Taking these facts into account, an accurate assessment of mobility of persons over 65 years of age is necessary to ensure medically and financially appropriate clinical care for patients following fall-related injuries. This includes reasonable prevention, treatment, and follow-up on discharge from the clinic. Validated assessment systems such as the Parker Mobility Score, Barthel Index, and EQ-5D-5L are already widely used in clinical practice to assess the mobility, activities and quality of life of older people. However, these assessment options are mainly based on the patients’ self-perception, and therefore can result in a biased clinical judgment followed by inappropriate medical treatment. For example, a significant discrepancy between self-assessment and objective tests was found for patients with chronic back pain [14] or neurological diseases [15]. Since most current medical decisions for the rehabilitation phase after fall-related fractures in older patients depend on subjective rating systems, the aim of this study was to identify differences between the self-assessment and external assessment of the mobility of people over 65 years of age. If the statements of the participants deviate from those of external persons, conclusions can be drawn for the practical use of the respective scores. The results make it possible to formulate recommendations for dealing with this group of people in a clinical setting. Furthermore, the aim was to determine whether there are differences in the case of a misjudgment depending on the living situation and gender. It should be determined whether the cognitive state has an influence on the misjudgment, and whether a misjudgment leads to an increased risk of falling. Further possibilities for objectifying the subjective statements of patients should be mentioned.

## 2. Materials and Methods

Ethics statement: This study was registered and approved by the local ethics committee (project number: 18-212). Approval was given before the enrollment started. The study followed the declaration of Helsinki.

Study setup: This study was based on an anonymous and standardized questionnaire that consisted of already existing and validated clinical scoring systems (see Appendix A). A study assistant interviewed 222 participants over 65 years of age who were living in a nursing home or independently at home. In each case, an external person (one close relative each for independently living or one professional caregiver for one or more people in nursing homes) was asked for the external assessment. Care was taken to ensure that the external assessment was made without knowledge of the self-assessment, in order to exclude any possible influence of bias. Exclusion criteria for the study were a previously diagnosed dementia or attachment to a wheelchair. The selection of independently living participants and nursing homes was carried out via internet research and personal contacts of the study staff. The preparations for the interviews differed depending on the living situation of the participants. People who lived independently in their own home were contacted by telephone. They were informed about the content of the study and asked to consent to participate. For the interviews in nursing homes, the first contact was made via the management. The study was explained and permission to conduct the interviews was asked. If they were willing to participate, further organization was discussed with the management, who selected the test subjects according to the inclusion criteria and informed and enlightened the residents about the study. To create representative results for a large city and suburban region, the interviews took place in nursing homes and study participants’ homes, in and around Munich, Germany between April 2018 and April 2019. All study participants signed a written declaration of consent. All data is anonymized and cannot be assigned to a specific person.

Parameters and scoring systems: To describe the study population, data on gender (female, male), living situation (independent, nursing home), age and height were collected. With the help of the Body Composition Monitor (BF511, OMRON, Mannheim, Germany), in addition to body weight and BMI, the percentage of body fat, visceral fat and muscle mass was determined for willing participants.

The Parker Mobility Score was used to assess the quality of mobility. This score records existing mobility restrictions and was originally developed to describe functional deficits before and after a fracture. It consists of three questions about mobility: ability to get about the house, ability to get out of the house, and ability to go shopping. Depending on the extent of the mobility impairment, zero points are given for “not at all”, one point for “with help from another person”, two points for “with an aid”, and three points for “no difficulty”. This gives a maximum score of nine points, which reflects no mobility restriction [16]. The study participant and external person both answered for this score.

The Barthel Index was used as an evaluation method for the systematic recording of activities of daily life. It was developed in 1965 by Florence l. Mahoney and Dorothea W. Barthel. The index provides information about the independence or need for care of people and is nowadays used in the context of care assessment, especially in geriatrics and rehabilitation. Ten different activities are listed. These include feeding, bathing, grooming, dressing, bowels and bladder continence, toilet use, transfer, mobility, and climbing stairs. Depending on the activity, between 0, 5, 10, or 15 points can be achieved. The total score shows the degree of restriction. The maximum score is 100 points, which reflects complete independence. A result of 0 points corresponds to a complete need for care [17]. The study participant and external person both answered for this score.

The EQ-5D-5L questionnaire was used to measure the quality of life. This widely used questionnaire was developed by the EuroQol group in 2009 in order to describe and investigate health-related quality of life. Besides a visual analogue scale, restrictions are recorded in five dimensions: mobility, self-care, usual activities, pain/discomfort, and anxiety/depression. For each dimension, there are five different answer options: no, slight, moderate, severe, and extreme problems/unable. The responses are used to generate a health state profile that can be converted into a health state index score. A high index value stands for a subjectively perceived high quality of life [18,19]. The study participant and external person both answered for this score.

The cognitive state of the participants was evaluated with the Mini Mental Status Test by Folstein et al., on the one hand to check the exclusion criterion of dementia by uncovering possible cognitive deficits directly in the beginning of the questionnaire, but after the collection of the demographics, and on the other hand to identify a possible influence on the mobility assessment. This psychometric test is often used to clarify cognitive performance disorders. The test consists of different tasks to test cognitive skills. A maximum of 30 points can be achieved, which means that there are no cognitive restrictions. For this study, a limit value of 24 points was defined as the minimum score the participants had to reach in order to participate, this applied for all participants [20,21].

Using various risk factors from the Hendrich 2 Fall Risk Model, the participants’ risk of falling was assessed. This made it possible to determine whether an incorrect assessment of mobility was associated with an increased risk of falling. The risk factors are rated with different points, which are added in the end. The higher the score, the higher the risk of falling [22]. This score was only answered by the study participant.

*Statistical Analysis*: Study data was collected and managed using REDCap (University of Vanderbilt, USA) [23]. Sample size calculation was based on a significance level of 0.05 and a power of 0.8. Initially, the data were divided into female and male groups, as well as independent and nursing home groups. For the descriptive statistics, the mean, standard deviation, median, and range were calculated. In order to capture possible differences between the self- and external-assessment, we subtracted the value of external assessment from the self-assessment. Thus, a positive result reflects an overestimation and a negative value an underestimation. The statistical significance was calculated after determining a Gaussian distribution using the Wilcoxon rank-sum test (GraphPad Prism, USA) and assumed at a *p*-value of ≤0.05. The correlations between the data of the Parker Mobility Score, Barthel Index, and Mini Mental Status Test as well as the Hendrich 2 Fall Risk Model were examined by determining the Spearman correlation. The level of significance was set at *p* ≤ 0.05.

## 3. Results

### 3.1. Study Population

From 222 participants, 73.0% were female and 27.0% were male. Approximately, 60% of the participants (female: 58.6%, male: 60.0%) lived in nursing homes. Study population details are provided in Table 1.

### 3.2. Increasing Age Leads to Misjudgment of Mobility by the Parker Mobility Score

Self-assessment of mobility by validated questionnaires is often utilized to determine the rehabilitation program for geriatric patients after fall-induced fractures. However, as this self-assessment is solely subjective, and thus might lead to an underestimation or overestimation of mobility, we ask participants and their relatives or their caregivers the same questions.

As expected, women as well as men who live independently at home achieved high scores, while scores among participants in nursing homes vary widely with a tendency towards lower scores (Figure 1A). A more detailed evaluation of the difference between self and external assessment revealed that women and men both over, and underestimate their situation of mobility (Figure 1A, red and blue dots). While men who live independently tend to assess themselves correctly, the degree of misjudgment increases especially among women in old age in nursing homes (Figure 1B). Self-assessment indicated a significantly higher Parker Mobility Score, when compared to an external assessment for females over 65 years of age in nursing homes (*p* ≤ 0.01). The other evaluated differences showed no statistical significance.

### 3.3. Nursing Home Residents Tend to Overestimate Themselves with the Barthel Index

Women and men who live independently at home achieved a high score in the Barthel Index, whereas scores from participants in nursing homes vary greatly, as expected (Figure 2A). With a few exceptions, the independently living participants assessed themselves correctly. A closer look at the participants in nursing homes revealed that women and men both tend to overestimate their situation in activities of daily life (Figure 2A,B). While the overestimation in men is nearly linear regardless of age, the spread of overestimation in women increases with age (Figure 2B). Self-assessment indicated a significantly higher Barthel Index, when compared to an external assessment for females (*p* ≤ 0.01) and males (*p* ≤ 0.01) over 65 years of age in nursing homes. The other evaluated differences showed no statistical significance.

### 3.4. Geriatric Participants Have a High Subjective Well-Being

High EQ-5D-5L Index values are found in participants living independently and those in nursing homes with no significant gender-specific differences (Figure 3A). However, lower values are more pronounced in nursing homes. Independent of age, participants clearly overestimated themselves with a few exceptions (Figure 3A,B). Self-assessment indicated a significantly higher index value, when compared to external assessment, for females over 65 years of age living independently at home (*p* ≤ 0.01) and in nursing homes (*p* ≤ 0.01). Likewise, for males over 65 years of age living independently at home (*p* ≤ 0.01) and in nursing homes (*p* ≤ 0.05).

### 3.5. Correlation of Mobility Misjudgment and Cognition or Risk of Falls

As the participants’ cognitive state might have an influence on the self-assessment of mobility, we correlated the results of the Mini Mental Status Test to the difference of the Barthel Index and Parker Mobility Score. For women in nursing homes, there was a slight negative correlation between the cognitive ability and the difference of the Barthel Index (*p* ≤ 0.05), indicating that increase of cognitive impairment leads to a higher misjudgment of daily activities. In all other cases, no significant correlation could be demonstrated (Figure 4A,B).

In order to determine the extent to which an incorrect assessment is associated with an increased risk of falling, the correlation between the Hendrich 2 Fall Risk Model and the Barthel Index or Parker Mobility Score was determined. With the exception of women in nursing homes, the difference of the Barthel Index showed a slightly positive correlation between an increased risk of falling and a misjudgment (*p* ≤ 0.05) (Figure 5A). However, this increased risk of falling in the event of a misjudgment in the Barthel Index could not be demonstrated in women in nursing homes, who make up a large part of the entire study population (Figure 5A). Furthermore, no significant correlation could be demonstrated between the difference of the Parker Mobility Score and the Hendrich 2 Fall Risk Model (Figure 5B).

## 4. Discussion

Restoring mobility and functional independence after a fall is essential and a main goal of orthogeriatric co-management [12]. Geriatric scoring systems provide information about mobility status and enable conclusions about health status and quality of life. This information should be used to organize the best and most personalized care, aftercare, and prevention. However, the statements of patients in these scoring systems may differ from reality. The aim of this study was to uncover differences between the self-assessment and external assessment of the mobility of people over 65 years of age. With few exceptions, this study indicates that these persons tend to overestimate their situation in terms of mobility, activities of daily life and quality of life, when using the described geriatric scoring systems. Significant differences between self- and external assessments were found especially among women and men in nursing homes. This circumstance is of particular importance as, when using geriatric scoring systems in the context of the patient’s medical history, care must be taken to ensure that the correct information about the functional and mobility status is guaranteed. However, all questionnaires show natural limitations in terms of self-assessment of the interviewed patients.

The Parker Mobility Score is described in the literature as a convincing tool for evaluating mobility and widely used for orthopedic and orthogeriatric patients [24]. Although many participants over- or underestimated their mobility, a significant difference in the assessment could only be demonstrated in older women in nursing homes. Thus, these women should receive special attention when assessing their mobility with the Parker Mobility Score. For the other groups of participants this score can be recommended in order to assess their mobility. In the elderly, sarcopenia can also be a cause of restricted mobility. This can be assessed well in a further step, for example with the SARC-F-Score [25].

The study by Liem, I.S. et al. concluded that the Barthel Index is the most appropriate tool for evaluating activities of daily life [24] and can be used on people living independently at home in order to assess their activities of daily life. However, special attention should be given to women and men living in nursing homes, as a significant overestimation in the Barthel Index was found for both groups.

The results of this study are particularly striking for the EQ-5D-5L. Both women and men living at home or in nursing homes showed a significant overestimation compared to the external statement. Apart from the problem of self-overestimation, it can nevertheless be deduced as a positive aspect that a high index value, which stands for a high quality of life, is at least subjectively confirmed by the participants. However, there are some aspects that need to be considered. On the one hand, the EQ-5D-5L enables more differentiated answer options with five answer levels, each in contrast to the other utilized questionnaires. On the other hand, it must be questioned how far an external person can correctly assess the participants’ subjective state of pain and fear. Family caregivers tend to overestimate health restrictions in less visible aspects, such as pain or anxiety and depression, whereas professional health caregivers tend to rate patients equally [26].

It is interesting to note that a slight cognitive impairment does not seem to play a role in an incorrect self-assessment. However, an exclusion criterion of this study was the result of less than 24 points in the Mini Mental Status Test. Therefore, no statement can be made about the extent to which the assessment changes in patients with significantly greater cognitive impairments. Tinetti et al. reported in its study of fall frequency evaluation and their reasons, that cognitive impairment is also an independent risk factor for fall events [27]. However, our study does not provide any indications that an incorrect assessment of mobility generally correlates with an increased risk of falling. Only an incorrect assessment of the Barthel Index showed a slightly positive correlation with a higher risk of falling.

In our study, it is assumed that a higher value of self-assessment represents a self-overestimation of study participants. However, a limitation of our study approach is the heterogeneity, and therewith subjectivity, of the external assessors of mobility. It might be that the external person underestimates the participants’ situation. Since an assessment should be objective, in particular by professional caregivers, we assume that a higher value of self-assessment equates to overestimation. In addition, in clinical practice, one will initially rely on verbal statements of the patient and, according to our study, the participants rated themselves partly better compared to the external statements. In order to treat patients correctly, we believe that it is not sufficient to rely solely on questionnaires and patient self-assessment. There should always be an objective geriatric assessment, which uses objective, standardized tests to get an unbiased picture of the patient and his individual situation. This is all the more important as people in nursing homes in particular overestimate themselves here, but a large number of falls take place in these facilities. Furthermore, it should be noted that an assessment of relatives—who may be emotionally dependent and have no professional competence—can differentiate from statements of professional caregivers in nursing homes [26]. The assessment may depend on the mental condition, which could be influenced by factors such as immobility after surgery or restrictions due to pain or medication. Due to the recruitment area of our study, participants in a large city and the surrounding area, we assume that the results are quite representative for large clinics with specialized orthogeriatric care.

A recent study by Oftendal et al. linked the number of daily steps to the mortality of older Australian adults [28]. Patients often overestimated their daily activity and walking time [29,30]. New opportunities in assessing mobility such as modern, automated measuring methods (wearables) can be a way to obtain objective data on patient mobilization and might objectify the medical decisions [31]. These techniques allow a feasible opportunity to capture the actual mobility over longer time periods and in the habitual environment of the patients [32]. However, the wearables and the algorithm for data extraction have to be fitted for the patient. Older adults are often slow walkers and thus, specific algorithms are necessary for these populations to measure a realistic mobilization and long-term changes, including behavior and activities of daily living [33]. Directly capturing patients’ mobility and daily behavior with medically-approved wearables will help to optimize future concepts for personalized therapy and rehabilitation, as well as the outcome of orthogeriatric patients.

## 5. Conclusions

The findings of our study highlight the importance of critically considering the mobility situation of people over 65 years of age, especially of (female) residents in nursing homes, as misinterpretation of their mobility performances might lead to insufficient treatment and aftercare. For this reason, it may be useful to complement self or external assessments with a more objective method, such as wearables, to assess mobility in a regular and/or continuous manner. In the end, manifested consequences of wrong self-perception, such as falls and/or fractures, must be evaluated in longitudinal cohort studies.

## Figures and Tables

**Figure 1 medicina-57-00980-f001:**
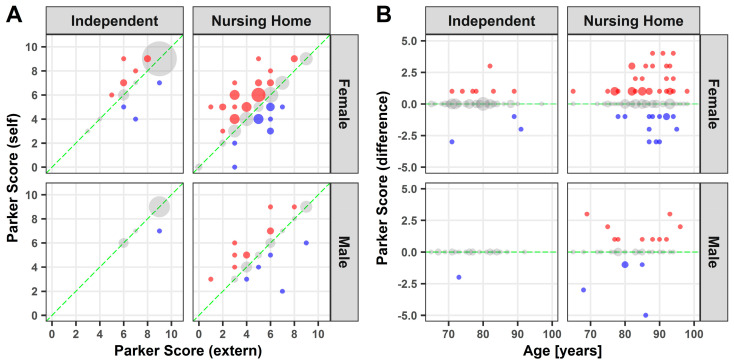
Parker Mobility Score: Point plot illustration of self and external assessment of the Parker Mobility Score (**A**) as well as the difference against age (**B**). Grey dots: no difference; blue dots: underestimation; red dots: overestimation. Dot size represents the relative number of participants.

**Figure 2 medicina-57-00980-f002:**
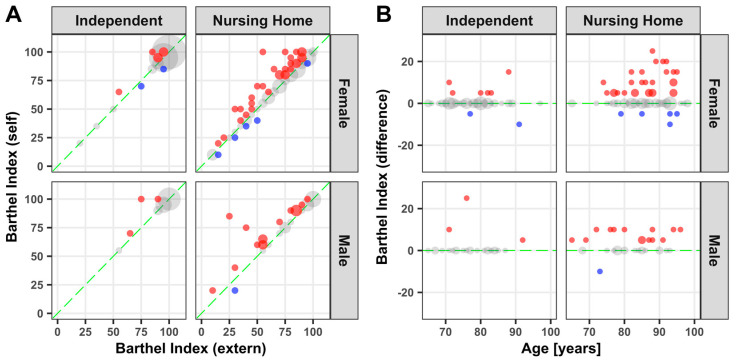
Barthel Index: Point plot illustration of self and external assessment of the Barthel Index (**A**) as well as the difference against age (**B**). Grey dots: no difference; blue dots: underestimation; red dots: overestimation. Dot size represents the relative number of participants.

**Figure 3 medicina-57-00980-f003:**
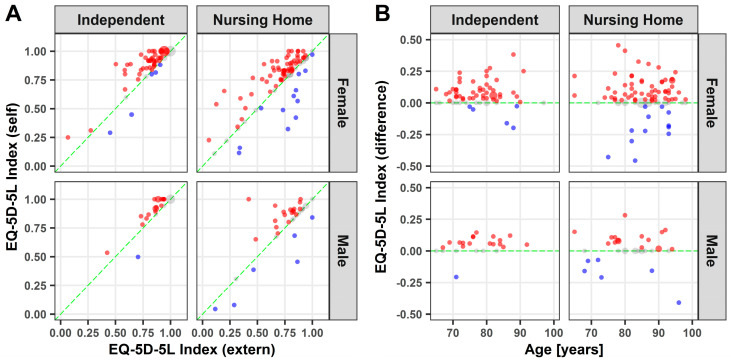
EQ-5D-5L Index: Point plot illustration of self and external assessment of the EQ-5D-5L Index (**A**) as well as the difference against age (**B**). Grey dots: no difference; blue dots: underestimation; red dots: overestimation. Dot size represents the relative number of participants.

**Figure 4 medicina-57-00980-f004:**
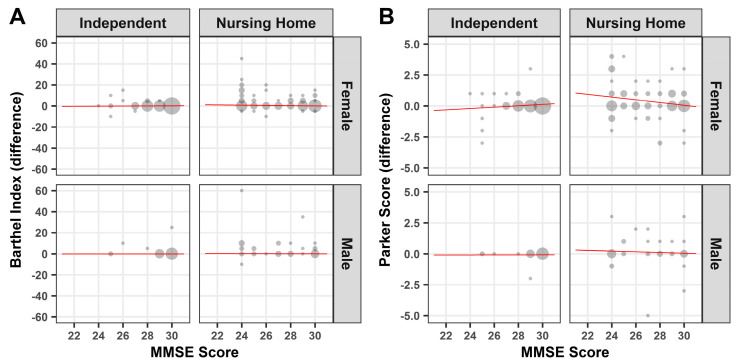
Mini Mental Status Test—Correlation: Point plot illustration of Barthel Index (**A**) or Parker Mobility Score (**B**) against the Mini Mental Status Test. Red line: linear regression.

**Figure 5 medicina-57-00980-f005:**
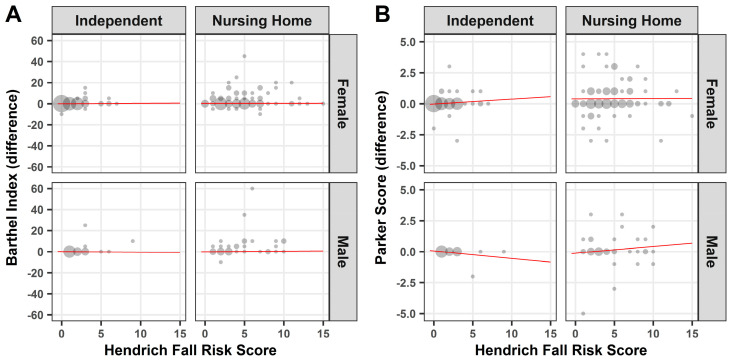
Hendrich Fall Risk Model—Correlation: Point plot illustration of Barthel Index (**A**) or Parker Mobility Score (**B**) against the Hendrich Fall Risk Score. Red line: linear regression.

**Table 1 medicina-57-00980-t001:** Geriatric study population. *n* = number of participants. n.s.= not significant. Median (Range).

	Female (*n* = 162)					Male (*n* = 60)				
	Nursing Home		Independent			Nursing Home		Independent		
	*n* (%)	Median (Range)	*n* (%)	Median (Range)	*p*	*n* (%)	Median (Range)	*n* (%)	Median (Range)	*p*
age (years)	95 (58.6)	86.0 (33)	67 (41.4)	78 (32)	*p* ≤ 0.01	36 (60.0)	83.0 (31.0)	24 (40.0)	76.5 (27.0)	*p* ≤ 0.05
height (cm)	95 (58.6)	163.0 (40.0)	67 (41.4)	164.0 (23.0)	n.s.	36 (60.0)	177.5 (29.0)	24 (40.0)	177.0 (26.0)	n.s.
weight (kg)		66.0 (101.0)		65.0 (48.0)	n.s.		80.0 (52.3)		79.2 (42.0)	n.s.
BMI		25.0 (31.6)		24.0 (18.5)	n.s.		25.7 (16.3)		25.9 (12.6)	n.s.
body fat (%)	13 (8.0)	34.6 (24.9)	34 (21.0)	35.3 (27.5)	n.s.	3 (5.0)	25.1 (11.5)	8 (13.3)	21.1 (11.7)	n.s.
visceral fat (%)		8.5 (11.0)		9.0 (11.0)	n.s.		6.0 (5.0)		10.1 (4.0)	*p* ≤ 0.05
muscle mass (%)		28.0 (8.1)		27.1 (22.0)	n.s.		33.1 (4.5)		33.2 (6.8)	n.s.

## Data Availability

The data presented in this study is available in the Appendix A.

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
