# Peer review of "Self-Assessment of Mobility of People over 65 Years of Age"

_medicina, 2021, doi:10.3390/medicina57090980_

Round 1

Reviewer 1 Report

The manuscript describes a technically sound piece of scientific research with data that supports the conclusions. The research has been conducted adequately.

Major concerns:

1. In the introduction, line (16-17) ”222 participants over 65 years and one external, closely related person each were interviewed by using a standardized questionnaire”, suggests that the external assessment was done uniformly by a single person, which was advisable and correct, especially if he/she was specialized in the medical assessment of patients.
In methods, lines 89-91, it is explained the reality of the assessment by a relative close to the patient (also subjective) or by the care staff who have more experience.
This heterogeneity and subjectivity of the assessment, which is recognized in the discussions, is good to be declared as a limitation of the study.

2.  Data related to women in nursing homes tend to be interpreted as minor, but there are still 95 participants out of 222 (more than 40%). 
For women in nursing homes, there was a slight 214 negative correlation between the cognitive ability and the difference of the Barthel Index (line 214-216)
With the exception of women in 222 nursing homes, the Barthel Index showed a slightly positive correlation between an increased risk of falling and a misjudgment (p ≤ 0.05)(line 222-224) Please elaborate.

Minor concerns

To better understand the graphs, please present as additional material in the form of tables all the data based on which you made the diagrams presented as Figure 1, 2, 3, 4, 5.

Reviewer 2 Report

The manuscript entitled „Self-assessment of Mobility of people over 65 years“ is interesting, useful and results are applicable in the elderly population. A different validated instruments are used in one place which indicate the necessary conclusions related to the elderly. It would be interesting to link whether BMI affects the risk of falls. Authors may consider showing that connection as well. 

Comments and suggestions can be found in the next section

  1. Introduction: authors should add some facts/ references from recent research according to statements of opinion on assessment and patient's self-perception in the last chapter of introduction (page 2, rows 62-80)
  2. The study setup: The authors should describe how they recruited respondents, especially those living in their homes with excluding criteria. The sample size and statistical power should be considered in this study.
  3. the authors should provide one or two sentences to explain why they chose the participants in and around Munich? Authors should describe the second group of respondents “closely related persons”. Can you show more clearly how many "close relatives" there were and how many "professional caregivers" in a particular group of respondents. Were there 222 of them too?
  4. The Parker mobility Score originally described functional deficits before and after the fracture. Was the number of previous falls important as an inclusion criterion?
  5. Which questionnaire did the respondents fill out first? I suppose that was MMS as one inclusion criteria. Authors should describe that.
  6. Results: in Tables 1 (geriatric study population) and 2 (Body parameters) shown results are a confusing. The author should present the "Study population" more clearly. I suggest merging two tables into one, with all the necessary parameters (n, %, age, BMI, body parameters), and the line "Total number of participants" should be added. The authors can compare these basic parameters according to age or find differences in BMI between participants living in home and those living in nursing homes.
  7. Chapter 3.2.: it is not necessary to be separated
  8. Conclusion from Abstract should be slightly expanded in a separated chapter of the manuscript
  9. Reference no 15. - correct
